# Defining harmful news reporting on community firearm violence: A modified Delphi consensus study

**Jessica H. Beard**[1,2]*, **Evan L. Eschliman**[3], **Anita Wamakima**[1], **Christopher N. Morrison**[3,4], **Jim MacMillan**[2], **Jennifer Midberry**[5]

**1** Department of Surgery, Division of Trauma Surgery and Surgical Critical Care, Lewis Katz School of Medicine, Temple University, Philadelphia, PA, United States of America, **2** Philadelphia Center for Gun Violence Reporting, Philadelphia, PA, United States of America, **3** Department of Epidemiology, Mailman School of Public Health, Columbia University, New York, NY, United States of America, **4** Department of Epidemiology and Preventive Medicine, School of Public Health and Preventive Medicine, Monash University, Melbourne, VIC, Australia, **5** Department of Journalism and Communication, Lehigh University, Bethlehem, PA, United States of America

* jessica.beard@tuhs.temple.edu

**Data Availability Statement:** Data cannot be shared publicly to protect participant anonymity. Data are available from the Temple University Institutional Review Board (contact via irb@temple.

## Abstract

Community firearm violence (CFV), including fatal and non-fatal shootings that result from interpersonal violence, disproportionately harms people from marginalized racial groups. News reporting on CFV can further exacerbate these harms. However, examining the effects of harmful news reporting on CFV on individuals, communities, and society is hindered by the lack of a consensus definition of harmful reporting on CFV. In this study, we aimed to define harmful reporting on CFV. We used a modified, three-round Delphi process to achieve consensus among diverse stakeholders. Round 1 sought to assess consensus on 12 potentially harmful news content elements for three levels of harm (individual, community, and society). Round 2 invited panelists to rate the severity of each news content element at each level of harm. Round 3 asked panelists to agree or disagree with the panel's median severity rating of each element at each level of harm. Twenty-one panelists were recruited from three expertise groups (lived experience of CFV, journalism practice, scholarship) and all panelists completed all three rounds. In Round 1, no negative consensus was achieved for any of the proposed news content elements. In Round 2, panelists assigned moderate to severe harm ratings for all but two news content elements, and median harm ratings for each element varied across the different levels of harm. In Round 3, panelists reported high levels of agreement for each harm rating at each level. This modified Delphi process yielded a definition of the 12 elements that comprise harmful news reporting on CFV and severity ratings of harm caused by each element at each level according to expert consensus. Future work will use these results to evaluate and intervene on harmful reporting on CFV. Reducing harm from reporting on CFV can help address this health disparity and support evidence-based approaches to this urgent public health issue.

edu) for researchers who meet the criteria for access to confidential data.

**Funding:** This research was funded by the Stoneleigh Foundation (https://stoneleighfoundation.org/) (JHB) and the National Institute on Minority Health and Health Disparities of the National Institutes of Health (JHB, CNM, JM) [grant number R21MD019088, https://www.nimhd.nih.gov/]. ELE is supported by the National Institute on Drug Abuse of the NIH [grant number T32DA031099, https://nida.nih.gov/]. The content is solely the responsibility of the authors and does not necessarily represent the official views of the Stoneleigh Foundation or the NIH. The funders had no role in study design, data collection and analysis, decision to publish, or preparation of the manuscript.

**Competing interests:** The authors have declared that no competing interests exist.

# Introduction

## Community firearm violence and structural racism

Firearm-related injury is an increasing threat to public health in the United States (US) [1–6]. Community firearm violence (CFV), defined as fatal and non-fatal shootings that result from interpersonal violence, disproportionately harms people from structurally marginalized racial groups and occurs predominantly in communities where racist policies and disinvestment have contributed to the presence of concentrated poverty [7–14]. These race and place-based disparities in CFV incidence are rooted in the social and structural determinants of health, including historic and ongoing structural racism [7,8,10,14]. Racialized societal structures in the US perpetuate CFV through contemporary policies and actions including mass incarceration, state sanctioned violence, food insecurity, and inequitable access to health, education, and social services, among others [15–17]. These structural forces shape an individual's risk for CFV and must be addressed to effectively decrease CFV incidence and promote community health and well-being [17].

## News reporting on community firearm violence in the United States

News coverage of CFV has the potential to deepen racialized structural inequities by shaping how the public and policy makers understand and respond to CFV [18–23]. In the US, news outlets rarely contextualize the complexity of CFV as an issue rooted in historic and contemporary structural racism [24–28]. Instead, news reports on violence typically emphasize individual blame over structural causes and have been critiqued for producing stereotypical narratives about Black people and communities [26,29–33]. The transmission of messaging about CFV—and its potential harms—depends in large part on how a news story is told, a concept called *framing*. Framing in news media occurs when journalists make choices about which content to include or exclude in a story, whose perspectives to highlight, and what attributes of an event to emphasize over others [27]. Those decisions make certain facets of a story more salient to audiences, and when the same messages are repeated in the news over time, people are likely to internalize those ideas [27,34,35].

A specific type of framing analysis that communication scholars have applied to news coverage of violence and crime is to distinguish between *episodic* and *thematic* framing [18,26,28]. *Episodic framing* of CFV is when a story focuses on a shooting event in isolation, while *thematic framing* places the event in its relevant social and structural contexts [18,24–26]. The primary source of a news report is also a key aspect of framing. In the case of CFV, most news reports rely on police sources and rarely include the perspectives of firearm-injured people and co-victims [26,28,36,37]. Decades of journalism and communication research have demonstrated that the *crime frame* is the dominant frame in news stories about violence [18,24–26,28,36]. *Episodic crime framing* is therefore news framing that is episodic, defines violence as a crime issue, and privileges police narrators above other viewpoints. Studies indicate that *episodic crime framing* in news reports about violence has detrimental societal level impacts; it can lead news audiences to blame individual victims, reinforce racist stereotypes about the people and places impacted, suggest an unfounded efficacy to policing as a means to prevent violence, and undermine effective public health responses [18,21,24–26,36,38,39].

Because of these theoretical and empirically demonstrated harms, scholars and journalism leaders have long advocated for reconsidering *episodic crime framing* in favor of other approaches that allow for more nuanced dialogue on complex issues like violence. One alternative frame posited by scholars and journalism educators for stories on CFV is the *public health frame*, which may include epidemiologic context and root causes along with public health

narrators, visuals, and solutions [20,22,25,28,40]. Additionally, the Solutions Journalism Network provides guidance to journalists on how to rigorously report on social problems like violence, highlighting the importance of including complex narratives and critical evaluation of solutions to create more impactful news stories [41].

In addition to a relatively large body of media effects research on news about violence and crime at the societal level, a few recent studies have examined the impact of news reports on individuals and local communities. For example, an interview-based study investigated the mental health consequences for Black participants of engaging with news media about police brutality against Black Americans [42]. Participants explained that such stories caused them to feel debilitating sadness, remain in a state of hyper-arousal and perpetual fear, and experience a sense of helplessness [42]. Another recent study, focused specifically on CFV, analyzed interviews with firearm-injured people and found that *episodic crime* narratives of participants' shootings made them feel dehumanized and added to their trauma [43]. Firearm-injured people in that study described how specific news elements, including graphic content, factual inaccuracies, and naming the treating hospital resulted in distress, harms to reputation, and threats to personal safety [43]. At the community level, some participants in the study expressed beliefs that harmful reporting may be driving increased CFV incidence in their neighborhoods by generating fear, which increases firearm purchasing and carrying [43]. Another community-level impact was observed in an online survey of Chicago residents, which found that participants who were more fearful of crime experienced higher levels of depression; however, this effect was dampened for individuals who paid closer attention to positive local news [44]. Taken together, these studies suggest that reducing harmful content elements in news coverage of CFV may ameliorate negative effects on individuals and communities faced with high rates of CFV [42–44].

## Defining harmful reporting as a next step for intervention

Events involving human suffering present journalists with the inherent tension of carrying out two key, but sometimes conflicting, principles of news: 1) to seek truth and report it and 2) to minimize harm [45]. Therefore, defining which specific content in CFV stories is harmful and developing reporting guidelines can illuminate (though not fully resolve) complicated newsroom debates about how journalists can best balance their duties to both inform the public and reduce harm. There is clear precedent for research exploring harmful news content to inform journalistic policy development as a public health intervention. With empirical support, journalistic guidelines that provide special instruction to minimize harm to victims and audiences cases of suicide, mass shootings, sexual assault, abuse, and crime involving minors have been widely accepted [45,46]. For example, studies demonstrating that harmful reporting approaches are associated with increases in suicide incidence led to the adoption of revised newsroom practices endorsed by public health experts [47–53]. These guidelines contain specific harm-reduction recommendations for media, including avoiding: prominent story placement, sensationalizing headline/content, glamorization or oversimplification of suicide, discussing the suicide method, and repeated reporting about the same suicide [47,49]. When media reports limit these harmful approaches, portray suicide as preventable, and provide resources, these modifications have been shown to result in population-level decreases in suicide rates [51–53].

Conceptualizations of harmful reporting on CFV remain far less developed than those that exist for reporting on suicide and mass shootings. Although there is clear evidence of negative societal level impacts of *episodic crime framing* and emerging research is uncovering the individual and community level harms of existing news coverage of CFV, guideline and

intervention development remains hindered by a lack of a consensus definition of harmful reporting on CFV [18,24,26,43]. Codifying the definition of harmful reporting on CFV through expert consensus is an essential next step to inform measurement of the extent and direct effects of harmful reporting in US news and to support the creation of evidence-based strategies for journalists to minimize harmful reporting on CFV. Because CFV largely impacts structurally marginalized people and places, harmful reporting on CFV also exerts disproportionate impact on people and communities subjected to systemic disadvantage. As such, harmful reporting on CFV may contribute to health inequities and perpetuate sustained disadvantage in a mechanism similar to other discriminatory practices and policies [8,10,14]. If this is true, then modifying news media approaches to limit harmful elements is a potential target for mitigating contemporary structural racism, minimizing disparities in CFV incidence and for helping CFV prevention efforts.

We utilized the social-ecological model of health as a framework to operationalize potential levels of harmful reporting on CFV [54–56]. We considered that individuals are nested within their social and physical environments at multiple levels (e.g., neighborhoods, community, society) and that harmful reporting of CFV may have adverse effects at each level [54–56]. The model presented in Fig 1 reflects these theorized levels and potential mechanisms of harm, informed by a review of previous research and work with our partners at the Philadelphia Center for Gun Violence Reporting (PCGVR), a community-based organization that supports ethical media reporting on CFV [18,24,26,38,39,43,57,58]. The three levels of harm corresponding to our social-ecological model considered throughout this study are defined as follows:

1. **Individual:** Firearm-injured people and/or co-victims involved in the shooting being reported on;

2. **Community:** Firearm-injured people and/or co-victims who have been affected by previous shootings;

3. **Society:** News audiences viewing, reading, and/or listening to the content and/or society at large.

## Study objective

In this study, we aimed to define harmful reporting on CFV using a modified Delphi method to achieve consensus among key expert-stakeholders identified in collaboration with community partners at PCGVR. We report on the Delphi process along with our findings about the specific news content elements that are potentially harmful to firearm-injured people, impacted communities, and society across our pre-defined three levels of harm (i.e. individual, community, and society).

## Materials and methods

### Panelists

We conducted this modified Delphi consensus study in collaboration with journalists and individuals with lived experience of firearm violence at PCGVR. PCGVR is a Philadelphia-based organization that supports "ethical, empathetic, and impactful" reporting on CFV through novel media created by firearm-injury survivors, educational programs for journalists, and community-engaged research [58]. We defined expertise on harmful reporting on CFV broadly and inclusively, working with PCGVR leadership to identify key stakeholders from three distinct expertise backgrounds: (1) lived experience experts (people who have been shot

**Fig 1. Social-ecological model of impacts of harmful reporting on community firearm violence (CFV).**

and survived as well as co-victims, who are the loved ones of firearm-injured people); (2) journalism practice experts (journalists with experience covering CFV); and (3) scholars (researchers of news reporting on violence). We aimed to include 20 panelists from these stakeholder groups, a larger sample size for the Delphi method to accommodate potentially varied perspectives [59]. Identifying nationally renowned academics and drawing on PCGVR's local network, we invited 25 potential adult panelists, including 9 lived experience experts, 8 journalism practice experts, and 8 scholars.

## Study design, procedures, and ethical statement

The Delphi method is a well-established and systematic process for iteratively developing consensus among a panel of experts [59–61]. This approach is especially practical where knowledge is "uncertain or incomplete, and human expert judgment is better than individual opinion" [61]. One of the key strengths of the Delphi process is panelist anonymity, which allows panelists to share their perspectives without conflicts, removing biases that might come from individual dominance or group conformity [61]. Other features of the Delphi method include controlled feedback by the research team and iterative discussions between rounds [61]. Following each round, the Delphi research team analyzes both quantitative and qualitative results, presenting these to panelists and using the results to develop the next round. We chose the Delphi method for this study for several reasons. Discussing firearm violence can be an emotional topic, especially for those with lived experience. The Delphi method allows panelists to share their perspectives anonymously ensuring that each perspective is considered and incorporated without risk of interpersonal disagreement. Additionally, because the topic of harmful reporting is highly complex, the Delphi method provides the opportunity to bring diverse experts into nuanced conversation that would not be achieved through a traditional survey or other quantitative research methods.

We designed this modified Delphi consensus study to include three rounds with a plan to stop the study after the third round [59,60]. No other stopping criteria were considered in the initial design. The study goal was to achieve expert consensus on what constitutes harmful reporting on CFV. In Round 1, the objective was to examine the degree of expert consensus on potential harmful reporting elements identified from prior research. In Round 2, the objective was to evaluate panelist perspectives on the severity of each potentially harmful reporting element that was not dropped due to negative consensus in Round 1. Following recommendations for Delphi consensus studies, the *a priori* objective of Round 3 was to allow panelists to evaluate the results of the previous rounds, revise their judgements, and specify any reasons for being outside consensus [59,60].

Potential panelists were contacted over email to introduce the study on October 31, 2023. They were provided a link to RedCap to complete Round 1, which took them to a written informed consent statement. Individuals who agreed to participate clicked "Yes" in response the consent question and then proceeded with answering the questions in Round 1. Basic demographic information was collected about each panelist along with information on their relevant expertise. The three rounds were carried out over approximately 5 months, allowing time between rounds for analysis, creation, and iteration of the subsequent rounds. The final round was completed on March 7, 2024. Panelists received $75 upon completion of each survey, for a total of $225 compensation for participation in all three rounds. Identifying information about the panelists, which was collected for the purposes of compensation, was stored separately from questionnaire responses to encourage open dialogue and responses and maintain anonymity. The Temple University Institutional Review Board approved this study.

## Round 1 development and analysis

We identified a set of potentially harmful news content elements informed by the prior qualitative interview study with firearm-injured people and by review of the journalism and communication literature about the harms of *episodic crime framing* in reporting on crime and violence [18,20,24,26,43]. This list of elements and corresponding descriptions was compiled by the first author (JHB), edited in two rounds with another author (JM) and then presented to the research team for feedback and in-team consensus. This approach ensured that the perspectives of people with lived experience of CFV were prioritized in the development of

**Table 1. Potentially harmful news content elements included in Round 1 of the modified Delphi consensus study.**

| Potentially Harmful News Content Element | Description |
|---|---|
| Graphic content | News coverage includes graphic or explicit news content about firearm violence, such as a video of shooting or a detailed description of the crime scene. |
| Clinical condition | News coverage of a shooting includes information on the clinical condition of a firearm-injured person (e.g. "critical" or "stable"). |
| Number of gunshot wounds | News coverage of a shooting includes information on the number of gunshot wounds of a firearm-injured person. |
| Name of treating hospital | News coverage of a shooting includes the name of the treating hospital of a firearm-injured person. |
| Relationship between firearm-injured person and perpetrator | News coverage of a shooting includes information on the relationship between the firearm-injured person and the alleged perpetrator of the shooting. |
| Mugshot | News coverage of a shooting includes a mugshot of the alleged perpetrator |
| Absence of a follow-up story | There is no follow-up story (e.g. an update on how a community has fared after a shooting or an interview with a survivor about their recovery) after the initial "breaking news" coverage. |
| Episodic framing | News coverage of firearm violence that focuses only on a specific shooting event and does not include context, root causes, or solutions to firearm violence. |
| Only law enforcement narrators | News coverage of firearm violence that only or predominantly presents the perspectives of law enforcement representatives (e.g. police). |
| Missing perspective of firearm-injured person | News coverage of a shooting that does not include the perspectives of the firearm-injured person and/or their loves ones. |
| Missing community perspective | News coverage of firearm violence that does not include the perspectives of people from the impacted community. |
| Does not explore solutions | News coverage of firearm violence that does not explore potential solutions. |

content for Round 1 [43]. Potentially harmful news content elements included in Round 1 are summarized in Table 1.

Traditionally, Delphi consensus studies are conducted in four rounds, with the first round focused on open-ended questions and idea generation [59–61]. Because the existing research on harmful reporting on violence is relatively robust, we elected to conduct a modified Delphi study in three rounds. As such, Round 1 was mostly structured with some space for open-ended responses from panelists throughout [59,60]. A 7-point Likert scale was used to assess the extent to which panelists agreed or disagreed that each potentially harmful news content element could cause harm at each level of harm (i.e. individual, community, society) [54–56]. In all rounds, panelists were instructed to answer questions from their perspective based on their expertise in personal experience, professional experience, and/or research. Consistent with recommendations for Delphi studies, consensus was defined *a priori* according to the proportion of responses within a range [59,60]. For Round 1, we were most concerned with negative consensus, defined as 80% or more of the panelists responding "Disagree" (6) or "Strongly Disagree" (7) to the question of potential harm of each news content element on the 7-point Likert scale [59,60]. We planned to drop any news content element at any level of harm from subsequent rounds in the event of negative consensus.

Panelist demographic data were analyzed descriptively furnishing proportions, medians, and interquartile ranges. Kruskal-Wallis tests were used to examine differences in responses between expertise groups for each news content element and for each level of harm. Responses

to open-ended questions were analyzed via a pragmatic analysis approach that included inductive thematic coding by a single team member (ELE) and full-team discussion of responses and emergent themes [62–64]. When all team members met together to discuss the open-ended responses, collaborative consensus was employed in order to identify any potentially harmful news content elements or additional question blocks that could be incorporated into future rounds [65].

## Round 2 development and analysis

Given that no news content element from Round 1 achieved negative consensus, we assessed the severity of all potentially harmful news content elements included in Table 1 in Round 2. We asked panelists to rate the severity of potential harm for each news content element for each of the three levels of harm on an 11-point scale of 0 to 10. We provided the same detailed instructions to all panelists and provided descriptive labels on intermediate values (i.e., 0 indicated *no harm*, 5 indicated *some harm*, and 10 indicated *extreme harm*). Panelists were then asked to select the top three most harmful news elements to allow panelists to emphasize which elements they found especially harmful (e.g., in case they rated more than three elements as "extreme harm").

Based on the high endorsement of the harm of graphic content across all levels and the thematic analysis of open-ended responses from Round 1, additional questions were included in Round 2 to identify specific content elements panelists would consider graphic and/or explicit and to assess how the medium of the news report may impact the level of harm of graphic and/or explicit content. Twelve potentially harmful content elements were included, and panelists were instructed that this content could be present in video, audio, or still photographs. Panelists were also asked to rate graphic and/or explicit content of firearm violence presented in four potential formats (disturbing video, disturbing still photographs, disturbing audio, and disturbing detailed verbal description) on the same 11-point harm scale by level of harm. Open-ended questions were included to allow panelists to provide support for their ratings of harmful content elements, offer further examples of harmful graphic and/or explicit content, and add information not covered in Round 2.

We analyzed responses to Round 2 by calculating the median harmfulness score for each content element at each of the three levels of harm. The median scores for severity of harm of each content element at each level of harm were then categorized based on the definitions provided that corresponded to the 11-point scale. Scores of 8 to 10 were categorized as *severe harm*, 5 to 7 were categorized as *moderate harm*, and 4 and below were categorized as *mild harm*. As additional analyses, we calculated percentages of endorsement within the three harmfulness ranges and tabulated the number of times each element was mentioned in the top 3 most harmful news content elements. Differences in responses between expertise groups were examined using Kruskal-Wallis tests. Open-ended responses were analyzed thematically as previously described [62–65]. In this round, special attention was paid to organizing themes around explanations for harmfulness rating responses by level of harm and considerations related to graphic and explicit content.

## Round 3 development and analysis

At the beginning of Round 3, panelists were presented with a table summarizing the median harmfulness scores tabulated from Round 2 along with each elements' assigned severity rating (i.e. severe harm, moderate harm, mild harm). Panelists were given the opportunity to agree or disagree with the Delphi panelists' median rating. Panelists who disagreed were asked to provide their reasoning for opposing the presented rating. Positive consensus was defined

as ≥ 80% of panelists agreeing with the panel's median rating while negative consensus was defined as ≥ 80% of panelists disagreeing.

We calculated percent agreement with Round 2 median harmfulness ratings and assigned severity ratings for each potentially harmful news content element for each level of harm. We tested differences in rates of any disagreement and number of disagreements between expertise groups using Fisher's exact tests and Kruskal-Wallis tests, respectively. Open-ended responses providing rationale for disagreement with the harmfulness ratings were analyzed thematically to determine whether panelists outside consensus felt the harmfulness ratings should be more or less severe and to identify perspectives outside consensus [62,63,65].

The full content of Rounds 1, 2, and 3 are available in the (S1–S3 Appendices). All analyses were conducted using Stata 18 (StataCorp LLC, 2024), and results of statistical tests for differences between expertise groups are reported below when $p < .10$.

## Results

### Panelists

Of the 25 potential panelists invited, 21 individuals (84%) agreed to participate, and all 21 panelists completed all 3 rounds. The median age of the panelists was 48 years (IQR 33–61 years; range 29–75 years). Twelve panelists (57%) self-identified as Black, 2 (10%) as Latine, 6 (29%) as White, and 2 (10%) as Multiracial. Most of the panelists were women (n = 15, 71%). Six participants (29%) self-identified as lived experience experts, 12 were journalism practice experts (57%) and 6 (29%) were scholars. Of note, three panelists reported overlapping expertise: two panelists with lived experience and one scholar also self-identified as journalism practice experts. In the statistical tests for differences between expertise groups, these three panelists were classified based on their non-journalism expertise.

### Round 1

Table 2 presents the number and percentage of panelists who agreed (i.e., somewhat agreed, agreed, or strongly agreed) that the media content elements presented (Table 1) could cause

**Table 2. Agreement regarding harmfulness of potentially harmful news content elements by level of harm.**

| Potentially Harmful News Content Element | Individual, n (%)[a] | Community, n (%)[b] | Society, n (%)[c] |
|---|---|---|---|
| Graphic content | 21 (100) | 21 (100) | 20 (95) |
| Clinical condition | 15 (71) | 13 (62) | 10 (48) |
| Number of gunshot wounds | 20 (95) | 20 (95) | 16 (76) |
| Name of treating hospital | 16 (76) | 15 (71) | 9 (43) |
| Relationship between firearm-injured person and perpetrator | 17 (81) | 15 (71) | 12 (57) |
| Mugshot | 19 (90) | 16 (76) | 16 (76) |
| Absence of a follow-up story | 15 (71) | 14 (67) | 16 (76) |
| Episodic framing | 18 (86) | 18 (86) | 18 (86) |
| Only law enforcement narrators | 19 (90) | 19 (90) | 18 (86) |
| Missing perspective of firearm-injured person | 17 (81) | 17 (81) | 16 (76) |
| Missing community perspective | 17 (81) | 17 (81) | 18 (86) |
| Does not explore solutions | 19 (90) | 19 (90) | 20 (95) |

Numbers and percentages refer to panelists who somewhat agreed, agreed or strongly agreed that the news content element could cause potential harm.

[a]Individual defined as: Firearm-injured people and/or co-victims involved in the shooting being reported on.

[b]Community defined as: Firearm-injured people and/or co-victims who have been affected by previous shootings. [c]Society defined as: News audiences viewing, reading, and/or listening to the content and/or society at large.

harm to firearm-injured people, communities impacted by CFV, and society. Panelists had the highest proportion of agreement that the elements of graphic content (100%), number of gunshot wounds (95%), and mugshots, only law enforcement narratives, and not exploring solutions (all 90%) were likely to cause harm at the individual level. The same elements had the same highest proportions of agreement for harm at the community level, except for mugshots (76% for community level). The elements with the highest proportions of agreement for harm at the society level were graphic content (95%) and does not report solutions (95%). No potentially harmful news content element achieved negative consensus for any level, so all elements, at all levels, were advanced to the next round. There were no statistically significant differences in responses between expertise groups for any element at any level.

In the open-ended responses, panelists discussed perceived mechanisms of harm by news reports and identified other potentially harmful elements. Several panelists indicated that graphic content could re-traumatize people impacted by CFV, while publicizing the health status or treating hospital of a firearm-injured person could result in threats to privacy and safety. Panelists also identified factual inaccuracies as a mechanism of harm, which are known to be more present in reports that favor the perspectives of law enforcement over firearm-injured people and their loved ones. Finally, panelists suggested that episodic framing in news stories without community perspectives can perpetuate racist stereotypes about firearm-injured people through stigmatization and dehumanization. Other potentially harmful elements identified by panelists in open-ended responses centered on concerns regarding journalistic practice; these included interviewing without consent, a lack of diversity among journalists and narrators, and logistical challenges related to deadlines and access to data. Several panelists noted that "insensitive" images that perpetuate stereotypes along with sensationalized headlines and narration are potentially harmful news content elements.

## Round 2

The median severity scores by content element and at each level of harm endorsed by the panelists are presented in Table 3, along with the corresponding rating (i.e., severe for 10–8, moderate for 7–5, or mild for 4 or lower). The highest number of median severity scores of severe harm ratings were endorsed at the individual level (i.e., for firearm-injured people included in the report). The news content elements that had median severity score ratings of severe harm across two or more levels were graphic content, episodic framing, reports that do not include solutions, and reports that include only or predominantly law enforcement perspectives. Tests for differences in severity scores by expertise groups did suggest some differences for some elements at some levels. Reported severity for clinical condition at the society level ($p = .07$), number of gunshot wounds at the society level ($p = .03$), name of treating hospital at the society level ($p = .05$), mugshot at the community level ($p = .10$), episodic framing at the individual level ($p = .09$), and does not explore solutions at the individual level ($p = .02$) all appeared to be rated differently by expertise groups. In each of these instances, the scholar expertise group rated the lowest average harm.

Fig 2 summarizes the number of times each element was included in panelists' top three most harmful news elements for each level of harm. The elements most commonly included in panelists' top three most harmful news elements were graphic content (14 times at individual level, 14 times at community level, 10 times at society level), episodic framing (12 times at society level), only law enforcement narrators (10 times at society level), missing perspective of firearm-injured person (11 times at individual level), and does not explore solutions (12 times at society level).

Table 3. Median harmfulness scores of harmful news content elements by level of harm and severity ratings.

| News Content Element | Median Harmfulness Score (Rating)[a] | | |
|---|---|---|---|
| | Individual[b] | Community[c] | Society[d] |
| Graphic content | 10 (Severe) | 9 (Severe) | 8 (Severe) |
| Clinical condition | 7 (Moderate) | 5 (Moderate) | 4 (Mild) |
| Number of gunshot wounds | 7 (Moderate) | 6 (Moderate) | 5 (Moderate) |
| Name of treating hospital | 8 (Severe) | 5 (Moderate) | 3 (Mild) |
| Relationship between firearm-injured person and perpetrator | 5 (Moderate) | 5 (Moderate) | 5 (Moderate) |
| Mugshot | 7 (Moderate) | 6 (Moderate) | 6 (Moderate) |
| Absence of a follow-up story | 8 (Severe) | 7 (Moderate) | 7 (Moderate) |
| Episodic framing | 8 (Severe) | 8 (Severe) | 8 (Severe) |
| Only law enforcement narrators | 8 (Severe) | 7 (Moderate) | 8 (Severe) |
| Missing perspective of firearm-injured person | 8 (Severe) | 5 (Moderate) | 6 (Moderate) |
| Missing community perspective | 8 (Severe) | 7 (Moderate) | 7 (Moderate) |
| Does not explore solutions | 8 (Severe) | 8 (Severe) | 8 (Severe) |

[a]Severity ratings are: Severe, 10–8; Moderate, 7–5; and Mild, 4 and below.

[b]Individual defined as: Firearm-injured people and/or co-victims involved in the shooting being reported on.

[c]Community defined as: Firearm-injured people and/or co-victims who have been affected by previous shootings.

[d]Society defined as: News audiences viewing, reading, and/or listening to the content and/or society at large.

### Graphic content

Seven of the 12 specific graphic content elements were considered graphic and/or explicit by at least half of the panelists. These elements were: the uncovered body of a deceased firearm-injured person(s) ($n = 21$, 100%); blood at the crime scene or on objects ($n = 19$, 90%); the covered body of a deceased firearm-injured person(s) ($n = 18$, 86%); the actual shooting incident as it unfolds ($n = 18$, 86%); someone being loaded into or out of an emergency transport vehicle (e.g. ambulance, police car) ($n = 17$, 81%), family or friends crying about a firearm-injured person ($n = 13$, 62%), an injured survivor ($n = 11$, 52%). Panelists rated all formats of graphic content (i.e. disturbing video, still images, audio, and detailed descriptions of firearm violence) as severely harmful (median scores of 8 and 9) across all levels of harm (i.e. individual, community, society) except for "disturbing detailed verbal description of firearm violence" at the society level (median score = 7, moderate harm). A figure depicting the perspectives of the panelists on which specific news elements constitute graphic and/or explicit content as well as a table detailing the median harmfulness score and corresponding severity rating for each of the four formats of graphic content at each level of harm are provided in the (S1 Fig and S1 Table).

In the open-ended responses, panelists provided rationale for their assigned severity scores by level of harm. Multiple panelists indicated that not providing context or solutions could perpetuate negative stereotypes about firearm-injured people, emphasizing the importance of including these elements along with the need to center the perspectives of firearm-injured people, their loves ones, and community members in news reports on firearm violence to minimize bias. In considering harms at the societal level, some panelists noted that harmful content elements can perpetuate fear and reinforce violence, dehumanize firearm-injured people, and lead to "narrow-mindedness." Several panelists emphasized that reliance on law enforcement narrators encourages audiences to understand firearm violence as a "criminal legal" issue rather than a public health issue. In terms of specific open-ended responses related to graphic content, panelists added that the following elements could also be considered graphic content:

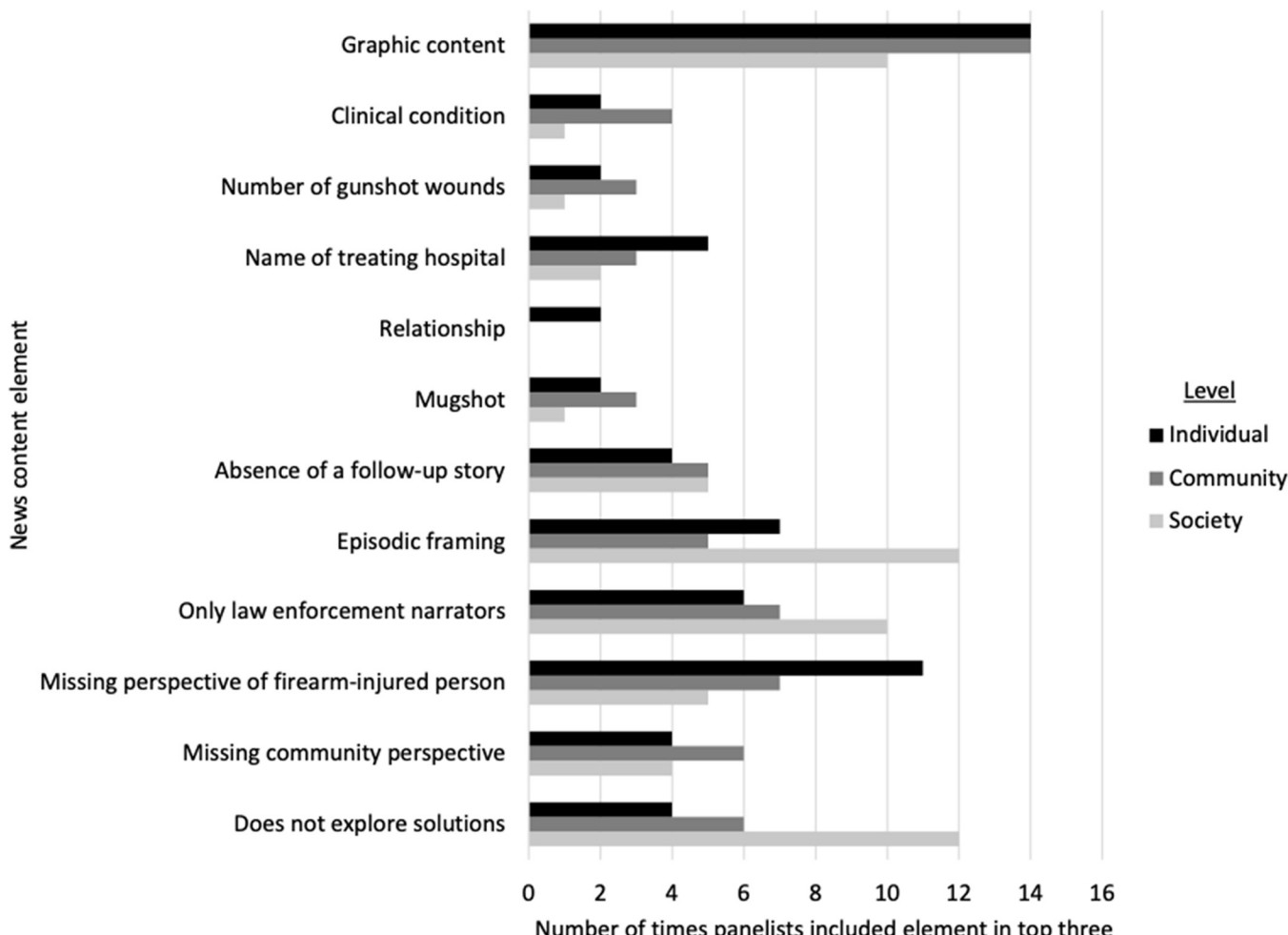

**Fig 2. Number of times element is mentioned in panelists' top three most harmful elements by level.** Relationship = Relationship between firearm-injured person and perpetrator. Individual defined as: Firearm-injured people and/or co-victims involved in the shooting being reported on. Community defined as: Firearm-injured people and/or co-victims who have been affected by previous shootings. Society defined as: News audiences viewing, reading, and/or listening to the content and/or society at large.

bullet holes in cars, audio of sirens or gunshots, stories about children being harmed, and "body counts" and/or statistics.

Several important tensions were highlighted by panelists in the open-ended responses. Some panelists noted that some information about firearm-injured people may be important to share in news reports to raise awareness about the issue and support the police in finding the perpetrator. They indicated that these considerations need to be balanced with maintaining the safety and privacy of the firearm-injured person and ensuring that the community's perspectives is centered in news reports about firearm violence. While the panelists uniformly agreed that graphic and explicit content were potentially harmful, a few panelists noted that some graphic content could also move people to action or prevent future shootings. Panelists noted that graphic and explicit content are more likely to cause harm to firearm-injured people, and that the recurring nature of graphic content in news reports on firearm violence likely results in desensitization of the public to firearm violence. One panelist suggested that a content warning ahead of graphic and/or explicit content might help reduce harm.

**Table 4. Percent agreement on Round 2 severity ratings for news content elements by level of harm.**

| Element | Percent Agreement (%, _n_ = 21) (Severity rating)[a] | | |
|---|---|---|---|
| | Individual[b] | Community[c] | Society[d] |
| Graphic content | 100 (Severe) | 100 (Severe) | 95 (Severe) |
| Clinical condition | 86 (Moderate) | 100 (Moderate) | 100 (Mild) |
| Number of gunshot wounds | 90 (Moderate) | 86 (Moderate) | 90 (Moderate) |
| Name of treating hospital | 90 (Severe) | 90 (Moderate) | 86 (Mild) |
| Relationship between firearm-injured person and perpetrator | 81 (Moderate) | 100 (Moderate) | 95 (Moderate) |
| Mugshot | 86 (Moderate) | 100 (Moderate) | 95 (Moderate) |
| Absence of a follow-up story | 90 (Severe) | 95 (Moderate) | 90 (Moderate) |
| Episodic framing | 100 (Severe) | 100 (Severe) | 100 (Severe) |
| Only law enforcement narrators | 100 (Severe) | 90 (Moderate) | 95 (Severe) |
| Missing perspective of firearm-injured person | 95 (Severe) | 95 (Moderate) | 95 (Moderate) |
| Missing community perspective | 95 (Severe) | 100 (Moderate) | 95 (Moderate) |
| Does not explore solutions | 86 (Severe) | 90 (Severe) | 90 (Severe) |

[a]Severity ratings reflect median harmfulness scores from Round 2 and are: Severe, 10–8; Moderate, 7–5; and Mild, 4 and below.

[b]Individual level defined as: Firearm-injured people and/or co-victims involved in the shooting being reported on.

[c]Community-level defined as: Firearm-injured people and/or co-victims who have been affected by previous shootings.

[d]Society-level defined as: News audiences viewing, reading, and/or listening to the content and/or society at large.

## Round 3

Table 4 summarizes the degree to which panelists agreed with the severity ratings resulting from the median harmfulness scores from Round 2. There was high agreement on the severity ratings, with all reaching positive consensus (i.e. ≥ 80% agreement). The lowest agreement was the rating of moderate harm for the element relationship between firearm-injured person and perpetrator at the individual level (81%).

Only 48 out of the 756 responses (6%) indicated disagreement with the panelist severity rating. Of the 21 panelists, 12 (57%) disagreed with a panelist severity rating at least once, and there was no difference between expertise groups in likelihood of disagreeing at least once ($p = .40$). The average number of disagreements was 2.3 (SD = 3.0, range: 0–10; median: 1), and there was no difference between expertise groups in number of disagreements ($p = .39$).

Thematic analysis found that 22 of the 48 disagreement open-ended responses reflected that the panelist felt the severity rating should be higher, while 26 reflected that the panelist felt the severity rating should be lower. In the disagreement open-ended responses, panelists discussed how the severity of harm from some news content elements (i.e., relationship between firearm-injured person and perpetrator, mugshot, and missing perspective of firearm-injured person) could be more or less severe depending on the circumstances of the shooting. Panelists also shared how they saw the harm of other news content elements (i.e., absence of a follow-up story and does not explore solutions) as depending on the nature of the reporting itself. For example, one panelist noted how covering solutions is only beneficial if the solutions covered are based in evidence or otherwise known to benefit firearm-injured people, communities affected by CFV, or audiences/society at large.

## Discussion

In this modified Delphi study, 21 panelists representing a diverse set of stakeholder perspectives and expertise, including people with lived experience of CFV, journalists, and scholars

determined through consensus that 12 specific news content elements are harmful to firearm-injured people, communities impacted by CFV, and society. Panelists agreed on a severity rating (mild, moderate, severe harm) for each news content element across three levels of harm (individual, community, society) and determined that news stories on CFV that include graphic content, episodic framing, and do not explore solutions have the potential to cause severe harm at all three levels of harm. In addition, panelists concluded that stories that only or predominantly include the perspectives of law enforcement have the potential to cause severe harm at the individual and society level and moderate harm at the community level, while news reports that neglect to include the perspectives of firearm-injured people and impacted community members were considered severely harmful at the individual level and moderately harmful at the community and society levels. Panelists found that harmful news content elements were most detrimental to firearm-injured people, with 8 of 12 elements rated as severe harm at that level. This is an especially important finding as traditional media effects theory and research largely considers the impact of news reports on news audiences more generally. These results highlight the need for journalists to consider the impacts of news reports on people directly impacted by CFV as a unique population requiring a trauma-informed approach to reporting [66]. Trauma-informed journalism employs methods that aim to recognize and respond effectively to the experiences of trauma survivors who are story sources or potential audience members with the goal to minimize further harm that the reporting may cause in the aftermath of a traumatic event [66–68]. Education and training in trauma-informed reporting practices for journalists is an important next step as journalists work to center the perspectives of survivors in news stories and minimize the news elements and practices known to cause harm to firearm-injured people [69].

This study has important implications for the study of news reporting on CFV. The level of consensus achieved indicates strong agreement regarding the potential for harm of the 12 specific news content elements among a diverse set of panelists. Panelists in this study agreed that reporting on CFV, which remains largely with *episodic crime framing*, is likely causing multi-level harms to firearm-injured people, their loved ones, communities impacted by CFV, and our society more generally [28]. This expert consensus provides strong rationale for the work of our community-based partners, PCGVR, in media narrative change and for this line of research examining harmful reporting as a health disparity that is contributing to the perpetuation of racial stereotypes and structural racism [57,58,69]. However, it is important to note that this study did not measure the direct impact of harmful reporting on individuals, impacted communities, or society, which remain important directions in future research. As a next step, we plan to use the results of this modified Delphi consensus to develop a novel instrument to measure harmful reporting along with a scale to document the severity of harm in news reports about CFV. We will then utilize this scale to examine whether there are racial and spatial disparities in harmful reporting on CFV. The ultimate goal of this line of research is to inform the measurement of direct effects of harmful reporting along with development and testing of interventions to minimize harmful reporting.

While the examination of the Delphi panelists' views on graphic and explicit content was not a stated goal of this study, our results do add a unique perspective to the rich and ongoing discussions surrounding the impact of news media portrayals of conflict and violence. Photo-journalists have traditionally argued that war photographs depicting human suffering can evoke compassion and spur action in audiences; however, research on the societal impact of violent images indicates that audience responses are highly complex [70–73]. While images that depict the human-cost-of war do have the potential to evoke compassion, the ubiquitous presence of US police killings of Black people in recent years in news and social media has caused societal-level trauma and negative psychological effects on Black Americans [72,73].

Compassion fatigue, feelings of helplessness, and the desire for media to cover resolution efforts are also potential responses to conflict images [71,74]. The results of our study uncovered similar tensions between the idea that graphic and explicit content can raise awareness of CFV as a social problem and the concern that such disturbing content can compound the trauma of firearm-injured people and co-victims. Despite this, panelists were conclusive that graphic and/or explicit content is potentially harmful across all levels of harm; 95% of the panelists endorsed that graphic and/or explicit content could cause severe harm to news audiences and/or society and they uniformly agreed that graphic and/or explicit content could cause severe harm to firearm-injured people and impacted communities. Research indicates that graphic coverage is not uncommon in news reports about CFV. A quantitative content analysis of television news in Philadelphia found that 7% of news stories on firearm violence in 2021 contained video or audio of a shooting event, which 86% of Delphi panelists considered harmful graphic content [28]. The results of this Delphi consensus can further urge journalists to refrain from including audio, videos, photographs, or detailed verbal descriptions of a shooting event in news stories on CFV to prevent harms to firearm-injured people, impacted communities, and news audiences. Or, when that is not possible, journalists may be able to mitigate harm by including a content warning ahead of graphic and/or explicit content in news reports on CFV.

This study has several strengths. All 21 panelists of this modified Delphi consensus study participated in all three rounds, indicating significant engagement and motivation among panelists in this study. While the study did achieve consensus on the severity ratings for all 12 news content elements, more than half of the panelists disagreed with at least one of the ratings. This suggests that while the consensus process was successful at synthesizing the viewpoints of the Delphi panelists, they felt free to disagree where they felt appropriate. One innovation of this study surrounds the definition of "expertise." Informed by the work of our community partners at PCGVR, we considered expertise on reporting on CFV to include not just academic scholars but also individuals with lived experience of CFV along with journalism practice experience. This inclusive perspective on "expertise," along with our engagement with community partners supports the validity of the Delphi panel findings. In this way, this research could serve as a model for other community-engaged efforts to achieve consensus on health or public health questions going forward.

The study findings should be interpreted considering its limitations. While we followed Delphi study guidelines, there are some methodological adaptations to note [59–61]. For example, it is possible that we missed news content elements that could be potentially harmful by commencing the study with a structured questionnaire. Additionally, while we were able to draw on panelists from across the US with academic and journalism practice expertise, our lived experience experts were identified in collaboration with our community partners at PCGVR and were thus mostly local to Philadelphia. While the objective of a Delphi panel is not necessarily generalizability, more lived experience experts from other areas of the country would have likely added depth to the panelist responses and our analyses. Also, the inclusion of more women than men in the study is a potential limitation. It is important to note the main goal of a Delphi panel is to create expert consensus where none currently exists. In this study, the aim was to achieve expert consensus on what constitutes harmful reporting on community firearm violence, which inherently relies on the perceptions and perspectives of the panelists. As such, we did not seek to measure the direct multilevel impacts of potentially harmful reporting on health and psychological outcomes, which will be an important area of research going forward. Importantly, while all elements included were indicated as potentially harmful in our prior qualitative research with people with lived experience of CFV, it is possible that further investigations of the impacts of these elements may not find quantitative

evidence of harms at some levels or may find quantitative evidence of harm at some levels but benefit at other levels.[43] Finally, this study did not evaluate the news content elements that should be present to support less harmful reporting, as there is far less existing research in this area. Thus, our results should not be interpreted to suggest that inclusion of firearm violence solutions is by definition "helpful," as news reporting about ineffective solutions without a critical lens could cause harm.

In conclusion, this study provides important and novel information about the elements that may constitute harmful news reporting on CFV during a significant national surge in CFV incidence in the US [1,2,4,6]. A Delphi panel with diverse expertise including lived experience of CFV, journalism practice, and academic scholarship reached consensus on 12 potentially harmful news content elements, rating the severity of harm attributed to these news content elements across 3 levels of harm (individual, community, and society). This study provides an important starting point from which to build scholarship around harmful reporting on firearm violence, work which will surely be iterative and evolve over time as the evidence base grows. The findings of this and subsequent studies will inform future investigations measuring the frequency and severity of harmful reporting on CFV in news reports, the direct impacts of potentially harmful news content elements across levels—and how these impacts may differ across levels—and in turn, inform interventions and best practices for journalists to minimize harmful reporting on CFV.

## Supporting information

**S1 Appendix. Round 1 content.**
(PDF)

**S2 Appendix. Round 2 content.**
(PDF)

**S3 Appendix. Round 3 content.**
(PDF)

**S1 Fig. Endorsement of specific graphic and/or explicit content elements as harmful.**
(TIF)

**S1 Table. Median harmfulness scores of formats for graphic and/or explicit by level of harm and severity ratings.**
(DOCX)

## Acknowledgments

We thank our panelists for providing their invaluable time and expertise to this study.

## Author Contributions

**Conceptualization:** Jessica H. Beard, Evan L. Eschliman, Christopher N. Morrison, Jim Mac-Millan, Jennifer Midberry.

**Data curation:** Jessica H. Beard, Evan L. Eschliman, Anita Wamakima, Christopher N. Morrison.

**Formal analysis:** Jessica H. Beard, Evan L. Eschliman, Christopher N. Morrison.

**Funding acquisition:** Jessica H. Beard, Christopher N. Morrison, Jim MacMillan, Jennifer Midberry.

**Investigation:** Jessica H. Beard, Evan L. Eschliman, Anita Wamakima, Christopher N. Morrison, Jim MacMillan, Jennifer Midberry.

**Methodology:** Jessica H. Beard, Evan L. Eschliman, Anita Wamakima, Christopher N. Morrison, Jennifer Midberry.

**Project administration:** Jessica H. Beard, Evan L. Eschliman, Anita Wamakima, Jim MacMillan.

**Resources:** Jessica H. Beard, Evan L. Eschliman, Christopher N. Morrison, Jim MacMillan, Jennifer Midberry.

**Software:** Evan L. Eschliman, Christopher N. Morrison.

**Supervision:** Jessica H. Beard, Christopher N. Morrison, Jim MacMillan, Jennifer Midberry.

**Visualization:** Evan L. Eschliman.

**Writing – original draft:** Jessica H. Beard, Evan L. Eschliman, Jennifer Midberry.

**Writing – review & editing:** Jessica H. Beard, Evan L. Eschliman, Anita Wamakima, Christopher N. Morrison, Jim MacMillan, Jennifer Midberry.

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
