## [Decision Letter · Decision Letter 0]

17 Jul 2024

PONE-D-24-20002Defining Harmful News Reporting on Community Firearm Violence: A Modified Delphi Consensus StudyPLOS ONE

Dear Dr. Beard,

Thank you for submitting your manuscript to PLOS ONE. After careful consideration, we feel that it has merit but does not fully meet PLOS ONE’s publication criteria as it currently stands. Therefore, we invite you to submit a revised version of the manuscript that addresses the points raised during the review process.

We look forward to receiving your revised manuscript.

Kind regards,

Claudio Terranova

Academic Editor

PLOS ONE

Journal Requirements:

2. PLOS requires an ORCID iD for the corresponding author in Editorial Manager on papers submitted after December 6th, 2016. Please ensure that you have an ORCID iD and that it is validated in Editorial Manager. To do this, go to ‘Update my Information’ (in the upper left-hand corner of the main menu), and click on the Fetch/Validate link next to the ORCID field. This will take you to the ORCID site and allow you to create a new iD or authenticate a pre-existing iD in Editorial Manager. Please see the following video for instructions on linking an ORCID iD to your Editorial Manager account: https://www.youtube.com/watch?v=_xcclfuvtxQ"

3. Thank you for stating the following financial disclosure:"This research was funded by the Stoneleigh Foundation (https://stoneleighfoundation.org/) (JHB) and the National Institute on Minority Health and Health Disparities of the National Institutes of Health (JHB, CNM, JM) [grant number R21MD019088, https://www.nimhd.nih.gov/]. ELE is supported by the National Institute on Drug Abuse of the NIH [grant number T32DA031099, https://nida.nih.gov/]. The content is solely the responsibility of the authors and dose not necessarily represent the official views of the Stoneleigh Foundation or the NIH."  

4. Thank you for stating the following in the Acknowledgments Section of your manuscript: "This research was funded by the Stoneleigh Foundation (JHB) and the National Institute on Minority Health and Health Disparities of the National Institutes of Health (JHB, CNM,JM) [grant number R21MD019088]. ELE is supported by the National Institute on Drug Abuse of the NIH [grant number T32DA031099]. The content is solely the responsibility of the authors and does not necessarily represent the official views of the Stoneleigh Foundation or the NIH."

Please remove any funding-related text from the manuscript and let us know how you would like to update your Funding Statement. Currently, your Funding Statement reads as follows: "This research was funded by the Stoneleigh Foundation (https://stoneleighfoundation.org/) (JHB) and the National Institute on Minority Health and Health Disparities of the National Institutes of Health (JHB, CNM, JM) [grant number R21MD019088, https://www.nimhd.nih.gov/]. ELE is supported by the National Institute on Drug Abuse of the NIH [grant number T32DA031099, https://nida.nih.gov/]. The content is solely the responsibility of the authors and dose not necessarily represent the official views of the Stoneleigh Foundation or the NIH."

5. In the online submission form, you indicated that data cannot be shared publicly to protect participant anonymity. Data are available upon request from the corresponding author for researchers who meet the criteria for access to confidential data.. 

Reviewers' comments:

Reviewer's Responses to Questions

**Comments to the Author**

1. Is the manuscript technically sound, and do the data support the conclusions?

Reviewer #1: No

Reviewer #2: Yes

2. Has the statistical analysis been performed appropriately and rigorously? 

Reviewer #1: Yes

Reviewer #2: Yes

3. Have the authors made all data underlying the findings in their manuscript fully available?

Reviewer #1: Yes

Reviewer #2: Yes

4. Is the manuscript presented in an intelligible fashion and written in standard English?

Reviewer #1: Yes

Reviewer #2: Yes

5. Review Comments to the Author

Reviewer #1: This paper raises some very important points about the role that media reporting can play in furthering the harms that are caused by firearm violence in communities. It is especially good to see attention being given to the disproportionate role that police sources play in shaping how news is framed. However, there is room for some improvement.

The paper currently lacks nuance, and makes many statements that draw artificial binaries – for example, that gun violence is framed as a crime issue when it should be seen as a public health issue. In reality, it is both of these, and far more – issues such as historical racism and its contemporary consequences, severe poverty, and socioeconomic marginalization go far beyond crime or public health. I would like to see a much more sophisticated theoretical and empirical perspective that takes this complexity into account.

Given that the entire purpose of the paper is to define ‘harmful’ reporting, it would be helpful to see more attention given to discussing current conceptualizations of, and evidence for, harm (or the lack of conceptualizations/evidence) and exploring how/why they differ or are unsatisfactory for research purposes. Further, there is a need to better explain whether and how the different ‘levels’ of harm were conceptualized/what guidelines were used to attempt to ensure some consistency between different individuals providing ratings – for instance, how can we be sure that different people in the different groups were using the same ‘yardstick’ to label something as an extreme level of harm?

The results may only be saying that some people/groups see things as more harmful than others – rather than telling us anything about actual harm caused.

This is particularly problematic in the context of lived experience participants, whose assessments could be influenced by a wide range of factors including trauma. This is touched on in the discussion, but only in the context of trauma-informed reporting (which means what, exactly?). The paper needs to be very careful about differentiating between subjective perceptions of harm, and whether those perceptions translate to harm in more ‘objective’ terms (for example, a person may believe that media coverage causes “narrow mindedness” – but is there anything to back this belief up?).

I am a little concerned about the overall objectivity and relative lack of serious critical analysis presented in the work. For example, something that is not really acknowledged is that, at times, an element that may be deemed ‘harmful’ may nevertheless be an important part of a balanced, objective, and accurate news story.

Similarly, some aspects that are implicitly considered to reduce harm – such as ‘providing solutions’ – may cause harm in their own right (e.g., ineffective ‘solutions’ may be promoted). Again, this speaks to the need for greater nuance and depth in how this paper is written.

I am concerned, in relation to the lived experience participants in particular, that the small sample size is unlikely to capture the breadth of different experiences people may have with media reporting. The over-representation of female participants is also a limitation.

Overall, the discussion takes an uncritical approach to the findings and accepts, seemingly without question, that the findings are valid, robust, and should be acted on. While clearly well intentioned, this comes across more as activism than scholarship. The paper also needs to be far more open about the limitations of the work.

Reviewer #2: Insightful and much needed article on harmful news content and gun violence reporting. I believe this manuscript will make significant contributions to our knowledge on the ways media content specifically gun violence is framed and in what ways we can collectively address harmful content. Overall the paper is well written. However, I do have several minor revisions. Author(s) should explain in more detail the Delphi method and why did the authors choose this approach. In regard to the recruitment of the sample the author(s) did not address: How were participants invited? If I am reading this correctly, a total of six participants with lived experience participated in study. What were their demographics? I am concerned with merging the sub-groups to arrive at a median age because most victims of violence are younger than 48. What is the average age of a gun violence survivor in Philadelphia this should be reflective in the age of the lived experience group. Furthermore, this sub-group should have been disproportionately over-represented than the other sub-groups in the study. The sample should be more reflective of people injured by firearm Philadelphia. If one of the aims is to center the voices of those most impacted by gun violence (which was evident by one of the categories), then the community voices should be over-represented in sample instead there was far more representation of journalists and scholars. This is a limitation of the study that should be acknowledged. Author(s) should also define: What does lived experience mean? Did participants witness violence, were they victims and/or perpetrators of violence?

6. PLOS authors have the option to publish the peer review history of their article (what does this mean?). If published, this will include your full peer review and any attached files.

Reviewer #1: No

Reviewer #2: No

---

## [Author Response · Author response to Decision Letter 0]

15 Aug 2024

Please see file "Response to Reviewers"

---

## [Decision Letter · Decision Letter 1]

9 Sep 2024

 PONE-D-24-20002R1 Defining Harmful News Reporting on Community Firearm Violence: A Modified Delphi Consensus Study PLOS ONE

Dear Dr. Beard,

Thank you for submitting your manuscript to PLOS ONE. After careful consideration, we feel that it has merit but does not yet fully meet PLOS ONE’s publication criteria as it currently stands. One of the reviewers, as you can read below, believes that not all the comments highlighted in the first review have been addressed. Therefore, we invite you to submit a revised version of the manuscript that addresses all the points indicated by the Reviewer.  Please submit your revised manuscript by Oct 24 2024 11:59PM. If you will need more time than this to complete your revisions, please reply to this message or contact the journal office at plosone@plos.org. Please include the following items when submitting your revised manuscript: A rebuttal letter that responds to each point raised by the academic editor and reviewer(s). You should upload this letter as a separate file labeled 'Response to Reviewers'.A marked-up copy of your manuscript that highlights changes made to the original version. You should upload this as a separate file labeled 'Revised Manuscript with Track Changes'.An unmarked version of your revised paper without tracked changes. You should upload this as a separate file labeled 'Manuscript'.

We look forward to receiving your revised manuscript.

Kind regards,

Claudio Terranova

Academic Editor

PLOS ONE

**Comments to the Author**

Reviewer #1: (No Response)

Reviewer #2: All comments have been addressed

2. Is the manuscript technically sound, and do the data support the conclusions?

Reviewer #1: Partly

Reviewer #2: Yes

3. Has the statistical analysis been performed appropriately and rigorously? 

Reviewer #1: Yes

Reviewer #2: Yes

4. Have the authors made all data underlying the findings in their manuscript fully available?

Reviewer #1: Yes

Reviewer #2: Yes

5. Is the manuscript presented in an intelligible fashion and written in standard English?

Reviewer #1: Yes

Reviewer #2: Yes

6. Review Comments to the Author

Reviewer #1: While this is an improved paper, there remain a few points that have not been adequately addressed.

Original comment: The over-representation of female participants is also a limitation.

Authors response: This was not included as a limitation because the gender of experts is not relevant to the validity of their expertise.

While this may be (somewhat) true of scholars and journalists, there is a growing body of research indicating that gender differences apply in how persons with lived experience respond to harm/emotion rating tasks, etc (which is unsurprising, given that these types of gender differences are found in general populations across a very wide range of socio-political contexts). Being a person who has experienced gun violence or a family member of a victim does not alter that. This in turn may have bearing on the results. This is a simple, reasonable, and evidence-based point to acknowledge.

Revised sentence: In this study, the aim was to achieve expert consensus on what constitutes harmful reporting on firearm violence, which inherently relies on the perceptions and perspectives of the panelists. As such, we did not seek to measure the direct impact of harmful reporting on health or psychological outcomes, which will be an important area of research going forward.”

This partially addresses my initial comment about the lack of objective information and potential mismatch between subjective and objective outcomes. However, this needs to be extended beyond health and psychological outcomes. Reporting may have psychological impacts for an individual. However, it may not have any negative broader community or societal impacts. Conversely, leaving an item out in order to avoid impacts on an individual may have negative community or social impacts. This requires more explicit acknowledgement (rather than just being pushed aside as a future direction). This also relates to the next point.

Initial comment: I am a little concerned about the overall objectivity and relative lack of serious critical analysis presented in the work. For example, something that is not really acknowledged is that, at times, an element that may be deemed ‘harmful’ may nevertheless be an important part of a balanced, objective, and accurate news story.

Authors response: This study was not meant to develop a list of things that should not be included, but instead to further advance research on this topic to inform guidelines and best practices to minimize any harms of reporting, which will inherently be iterative and evolve over time. We have now included statements to this effect, stating: “This study provides an important starting point from which to build scholarship around harmful reporting on firearm violence, work which will surely be iterative and evolve over time. The findings of this study will inform future investigations measuring the frequency and severity of harmful reporting on CFV in news reports, the direct impacts of harmful reporting on individuals and communities, along with interventions with journalists to minimize harmful reporting on CFV” (Page 23).

While this goes part of the way to addressing my concern, it still does not capture the key issue – specifically, that something that may be perceived by some as “harmful”, may nevertheless be helpful to others. The revised paper is still not acknowledging this issue. It is a very straightforward point to include, and not too much to ask.

Reviewer #2: The author(s) have addressed the necessary revisions, this paper is much needed in the gun violence reporting space.

7. PLOS authors have the option to publish the peer review history of their article (what does this mean?). If published, this will include your full peer review and any attached files.

Reviewer #1: No

Reviewer #2: No

---

## [Decision Letter · Decision Letter 2]

4 Dec 2024

Defining Harmful News Reporting on Community Firearm Violence: A Modified Delphi Consensus Study

PONE-D-24-20002R2

Dear Dr. Beard,

We’re pleased to inform you that your manuscript has been judged scientifically suitable for publication and will be formally accepted for publication once it meets all outstanding technical requirements.

Kind regards,

Claudio Terranova

Academic Editor

PLOS ONE

Additional Editor Comments

Regarding the issue raised in your cover letter dated October 4, concerning a comment by Reviewer #1, I have carefully reviewed both the revisions and the authors' responses. I believe there is a misunderstanding between what the reviewer intended in the request and what the authors appear to interpret.

As far as I can tell, the reviewer was simply asking the authors to acknowledge that there is a gender imbalance in their sample, and that this is related to a previous point (in the original review) regarding sample size, i.e., that this may not "capture the breadth of different experiences people may have." The authors' response is that "the gender of experts is not relevant to the validity of their expertise." However, I do not perceive the reviewers' concern to be that women lack expertise, but rather that their perspective may be over-represented.

In conclusion, I believe that the reviewer has not suggested that women lack expertise, and that the request to add a sentence noting that there were unequal numbers of men and women does not constitute sexism.

Therefore, I think the sentence you have introduced to address the reviewer’s comment can be maintained.

Reviewers' comments:

Reviewer's Responses to Questions

**Comments to the Author**

1. If the authors have adequately addressed your comments raised in a previous round of review and you feel that this manuscript is now acceptable for publication, you may indicate that here to bypass the “Comments to the Author” section, enter your conflict of interest statement in the “Confidential to Editor” section, and submit your "Accept" recommendation.

Reviewer #1: All comments have been addressed

2. Is the manuscript technically sound, and do the data support the conclusions?

Reviewer #1: Yes

3. Has the statistical analysis been performed appropriately and rigorously? 

Reviewer #1: Yes

4. Have the authors made all data underlying the findings in their manuscript fully available?

Reviewer #1: Yes

5. Is the manuscript presented in an intelligible fashion and written in standard English?

Reviewer #1: Yes

6. Review Comments to the Author

Reviewer #1: (No Response)

7. PLOS authors have the option to publish the peer review history of their article (what does this mean?). If published, this will include your full peer review and any attached files.

Reviewer #1: No

---

## [Editor Report · Acceptance letter]

8 Dec 2024

PONE-D-24-20002R2 

PLOS ONE

Dear Dr. Beard, 

I'm pleased to inform you that your manuscript has been deemed suitable for publication in PLOS ONE. Congratulations! Your manuscript is now being handed over to our production team.

Kind regards, 

on behalf of

Professor Claudio Terranova 

Academic Editor

PLOS ONE